# Thermal transport of helium-3 in a strongly confining channel

D. Lotnyk [1], A. Eyal[1,2], N. Zhelev[1], T. S. Abhilash [1,3], E. N. Smith[1], M. Terilli[1], J. Wilson[1,4], E. Mueller [1], D. Einzel[5], J. Saunders[6] & J. M. Parpia [1✉]

The investigation of transport properties in normal liquid helium-3 and its topological superfluid phases provides insights into related phenomena in electron fluids, topological materials, and putative topological superconductors. It relies on the measurement of mass, heat, and spin currents, due to system neutrality. Of particular interest is transport in strongly confining channels of height approaching the superfluid coherence length, to enhance the relative contribution of surface excitations, and suppress hydrodynamic counterflow. Here we report on the thermal conduction of helium-3 in a 1.1 $\mu$m high channel. In the normal state we observe a diffusive thermal conductivity that is approximately temperature independent, consistent with interference of bulk and boundary scattering. In the superfluid, the thermal conductivity is only weakly temperature dependent, requiring detailed theoretical analysis. An anomalous thermal response is detected in the superfluid which we propose arises from the emission of a flux of surface excitations from the channel.

[1] Department of Physics, Cornell University, Ithaca, NY 14853, USA. [2] Physics Department, Technion, Haifa, Israel. [3] VTT Technical Research Centre of Finland Ltd, Espoo, Finland. [4] SUNY Geneseo, Geneseo, NY 14454, USA. [5] Walther Meissner Institut, Garching, Germany. [6] Department of Physics, Royal Holloway University of London, Egham TW20 0EX, Surrey, UK. ✉email: jmp9@cornell.edu

Superfluid $^3$He under strong confinement provides a model system for the study of topological quantum matter and unconventional superconductivity. The two spin-triplet phases of topological superfluid $^3$He provide benchmarks for topological superconductivity[1], which has yet to be firmly established in a variety of candidate systems[2]. Confinement in a slab geometry is used to create a film of superfluid, with equivalent surfaces, of thickness comparable to the superfluid coherence length[3]. The distortion of the p wave order parameter can be directly measured by NMR[4]. This provides a model system to characterize surface pair-breaking in unconventional superconductors, in the absence of defect and impurity scattering, which is of relevance to future mesoscopic device applications of putative topological superconductors[5].

Surface pair-breaking, which can be tuned in situ[6], has a strong influence on the superfluid phase diagram; the chiral A-phase is favored over B-phase in extensive regions of the $p$–$T$ plane. Confinement has also been predicted to stabilize emergent phases, such as a spatially modulated superfluid[7]. A two-dimensional spatial modulation has recently been observed by NMR in superfluid $^3$He (ref. [8]). This is closely related to pair density waves that are under extensive investigation in unconventional superconductors[9].

In the chiral A-phase and time reversal invariant B-phase, distinct surface and edge excitations are predicted to emerge due to bulk-edge correspondence. These are expected to lead to spontaneous ground-state mass and spin currents[1,10,11]. The analogous phenomena have so far eluded observation in unconventional superconductors; for example, searches for edge currents arising from putative breaking of time reversal symmetry in Sr$_2$RuO$_4$ have not been successful[12,13]. The predicted surface and edge excitations are expected to be measurable in thermal transport[14–17] in $^3$He. Moreover, thermal analogs of the Hall effect are predicted in the $^3$He chiral A-phase and chiral superconductors[18,19] both, as a result of edge currents and scattering from impurities. Observation of these exotic phenomena in superfluid $^3$He requires confinement in precise geometries. However, as yet there are no measurements of the thermal conductivity of superfluid $^3$He under such confinement. Here, we report that thermal transport in confined channels is rich with unanticipated effects in both the normal and superfluid states. Quantifying this transport provides insight into the underlying kinetic processes and paves the way for distinguishing the signatures of topological superfluidity.

Experiments on $^3$He in the presence of disorder have shown that in addition to the modification of transport behavior from the pure liquid[20–23], superfluid phases unseen in the bulk emerge due to the anisotropy of the disorder[24–29]. These anisotropic structures have also led to the observation of half-quantum vortices[30,31]. On the other hand, nanofabrication techniques can be used to engineer anisotropic environments[32,33], with no accompanying disorder. More complex structures, such as channels or periodic arrays of posts, with typical length scales of a few coherence lengths, can also potentially tailor specific superfluid phases[33]. Thermal transport will play a key role in characterizing these tailored "materials" and hybrid structures. In the work presented here, the focus is on the understanding of thermal transport of $^3$He in a simple slab geometry with strong confinement corresponding to of order 15–50 times the pressure-dependent zero temperature coherence length, $\xi_0 = \hbar v_F/(2\pi k_B T_c)$ where $v_F$ is the Fermi velocity, $k_B$ Boltzmann's constant, and $T_c$ the superfluid transition temperature.

Thermal conductivity in normal liquid $^3$He is a diffusive process and can be understood in terms of the kinetic theory of quasiparticle excitations. Due to the Pauli exclusion principle, the phase space available for scattering becomes small at low

temperatures giving rise to a strong temperature dependence of the inelastic thermal mean free path, $\lambda_\kappa \sim T^{-2}$. This results in a bulk thermal conductivity, $\kappa$ that is proportional to $T^{-1}$ (since $\kappa = 1/3(C_v/V_m)v_F\lambda_\kappa$, where $V_m$ is the molar volume, $C_v \sim T$ is the molar specific heat, and $v_F$ is the Fermi velocity[34]). This behavior is observed in the bulk normal liquid, since, unlike other condensed matter systems, $^3$He is impurity-free and there are no elastic scattering centers. The introduction of a matrix of impurities (such as aerogel) modeled as a collection of point scatters, leads to a vanishing conductivity[35–37] as $T \to 0$ ($\kappa$ proportional to $T$), due to the mean free path being limited by scattering from the impurities. Recently, there has been significant renewed interest in hydrodynamic transport in electron fluids, arising from advances in materials[38]. Building on early work on two-dimensional electron gases in AlGaAs heterostructures[39], the required condition that electron–electron collisions dominate over electron–phonon or electron-impurity scattering is satisfied in ultraclean materials, such as graphene[40], PdCoO$_2$ (ref. [41]), and WP$_2$ (ref. [42]) leading to viscous and quasiballistic transport, with signatures distinct from ohmic transport. The confinement of such materials into restricted conduction channels is relevant for the understanding of the interplay of bulk and surface scattering, with many open questions. In this context, confined $^3$He, in which scattering between quasiparticles dominates in bulk, provides a useful paradigm, including the potential to crossover to quasi-two-dimensional transport[43] with strong confinement.

In our experiment, two chambers filled with bulk fluid, a small isolated volume, and a container with a heat exchanger through which the $^3$He is cooled, are separated by a nanofabricated 1.1 μm high channel. Both containers are equipped with tuning fork thermometers, which can measure the temperature or act as a heater. By injecting heat into one chamber and measuring the response, we explore the $^3$He diffusive thermal conductivity under strong confinement in both the normal and superfluid phase.

In the normal state, we find an anomalous thermal conductivity that is nearly temperature independent <10 mK, implying an effective mean free path that varies as $T^{-1}$. This is the same temperature dependence as that of the momentum relaxation time inferred from our earlier mass transport studies in $^3$He films on polished silver surfaces[43,44]. These results may be accounted for by quasiclassical interference between bulk scattering and that arising from surface disorder[45,46].

In addition to diffusive heat flow[47], superfluids support the thermal transport via a hydrodynamic process: two-fluid counterflow where relative motion of the superfluid and normal component results in heat flow. This effect is well-established[48–52] in studies of superfluid $^4$He, but results on superfluid $^3$He are limited[53,54]. In steady-state thermal counterflow through a channel, the temperature gradient generates a fountain pressure, such that the difference in chemical potential between the two ends of the channel is zero. The superfluid component (driven by gradients in chemical potential) flows at constant velocity toward the hot end. The fountain pressure forces the normal (entropy carrying) component in the opposite direction, with volume flow rate determined by viscous transport in the channel. If the normal component is not viscously clamped, this hydrodynamic thermal transport dominates and the thermal conductivity increases sharply as $T_c$ is traversed from the normal state[53,54].

One motivation of the present experiment was to quantify thermal transport under strong confinement in the superfluid phase, arising from quasiparticle excitations, by reducing thermal counterflow. Informed by prior mass flow studies[43,44,55], the strong confinement imposed by the 1.1 μm channel was designed to clamp the normal component even in the extreme Knudsen regime. The Knudsen regime onsets when the viscous mean free

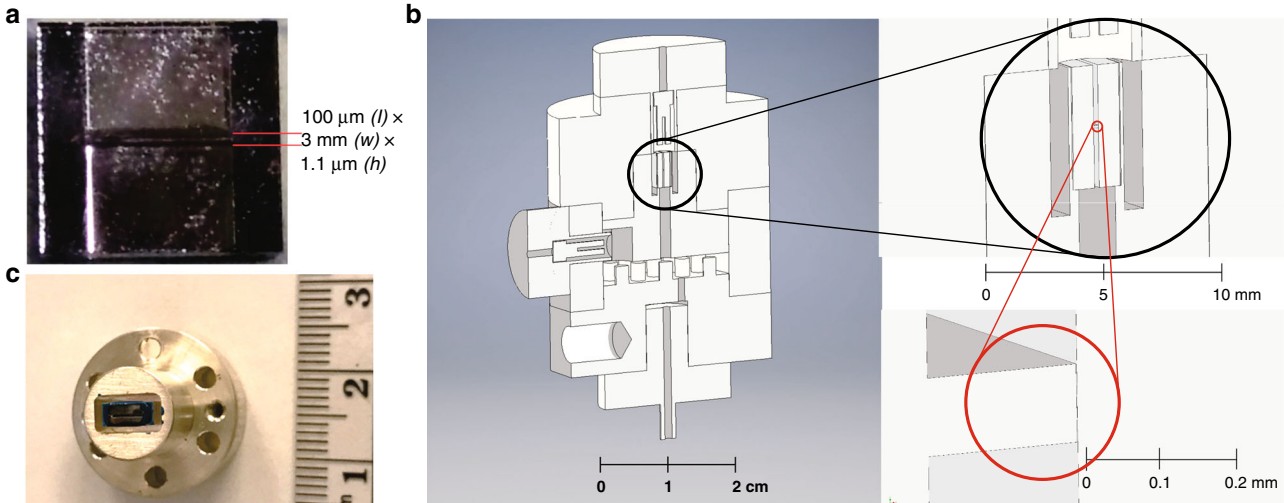

**Fig. 1 Experimental details. a** Image of the cavity containing the channel prior to mounting. **b** Schematic of the experimental cell with forks mounted in the isolated chamber (IC, depicted at the top) and in the heat exchanger chamber (HEC, located in the chamber below the channel) separated by the thermal conductance channel. The cavity in its mount is depicted schematically in the large black circle and the channel is depicted in the red circle. **c** Cavity mounted in coin-silver carrier, where the (blue) epoxy joint to the thin coin-silver wall is visible. Each small division is 1 mm.

path exceeds the height of a confining channel and is preceded by slip (the phenomenon where the velocity of a fluid in contact with a wall is nonzero) that allows the viscous fluid to move relative to the wall. Extrapolating the effective viscosity from work on a 135 μm channel[55] to the present 1.1 μm channel resulted in a prediction that hydrodynamic transport would be negligible (Supplementary Note 1). However, the influence of even limited slip-induced normal flow may be non negligible (as we will encounter in the following paragraph and in the discussion section), apparently resulting in the flow of surface excitations driven by the fountain pressure under these conditions of extreme confinement. In our experiment, we find that the measured thermal conductivity (in the superfluid state) shows a weak temperature dependence similar to theoretical predictions for the diffusive contribution to thermal conductivity in bulk. A full quasiclassical theory calculation of thermal transport in a strongly confined channel subject to both a temperature and fountain pressure gradient is required to fully understand thermal conduction in this regime.

Furthermore, in the superfluid state, we observe a rapid and unexpected response of the thermometer in the heat exchanger volume. The nonlocal response is indicative of transport of energy by excitations that are out of equilibrium with the bulk liquid. Although the overall length of the channel (including lead-ins) and the distance between forks is several mm, much longer than the inelastic mean free path of bulk quasiparticle excitations, the characteristics of the observed response indicate a ballistic flow of excitations, following the limited flow of the normal component induced by the fountain pressure created in the heated isolated volume. We propose that the excitations responsible for this transport are likely surface bound[56–59], and once detached from the surface interact only weakly with excitations in the intervening bulk fluid[60], and thus can travel long distances.

## Results

**Experimental details**. This paper describes the measurement of heat transport through a 1.1 μm high, 3 mm wide, and 100 μm long channel with 200 μm tall × 3 mm wide × 2.45 mm long "lead-in" sections at either end. The 1.1 μm height section should dominate the thermal impedance and the structure is shown in Fig. 1a, b. Two chambers sit on either end of this channel, one of which is

thermally anchored to the nuclear demagnetization stage through a sintered silver heat exchanger. We refer to this as the heat exchanger chamber (HEC; Fig. 1b). The second chamber (designated as the isolated chamber, IC) was cooled through the thermal impedance provided by the channel, or, at temperatures above ~20 mK, by direct thermal contact with the coin-silver walls via the Kapitza thermal resistance[61]. The channel (Fig. 1a) was nanofabricated in 1 mm thick silicon, capped with 1 mm thick sodium-doped glass, anodically bonded to the silicon[62]. The channel was glued into a coin-silver carrier (Fig. 1c) using TraBond 2151 epoxy. The temperature in each chamber was determined by a quartz "tuning fork" thermometer operating at 34 kHz.

A heat pulse was applied to the liquid in the IC by increasing the drive voltage applied to the fork in the IC by up to a factor of 10 for a period of 10–100 s in the superfluid and 60–300 s in the normal state. These pulses deposited energy of order a few nJ compared to the ambient power dissipated by the fork of order 0.1 pW. The drive was then restored to the usual level and the quality factor of the fork, $Q$ (and hence the temperature of the IC) was monitored through its recovery to determine the thermal relaxation time. The measured IC fork thermal relaxation time, $\tau$ was then related to the thermal resistance, $R_{th}$ through $\tau = R_{th}C$, where $C$ is the heat capacity of the $^3$He in the IC volume. The heat capacity was determined from the known specific heat of $^3$He (ref. [63]), and the calculated volume of the IC (0.14 ± 0.02 cm$^3$), shown in Fig. 1. The surface area of the IC was calculated to be 14.5 ± 0.5 cm$^2$, including the area of all metal surfaces wetted by $^3$He. By comparison, the HEC had a volume of 0.72 ± 0.1 cm$^3$ and a surface area of 3.5 ± 0.5 m$^2$. The geometry was chosen so that the anticipated thermal relaxation times[34,63] would lie between 100 and 3000 s, compatible with the response time of the tuning fork thermometer. The equilibrium temperature of the $^3$He sample was also determined by monitoring the $Q$ of the HEC tuning fork.

Data were obtained while warming and cooling the nuclear demagnetization stage to which the cell was thermally anchored. To compensate the ambient heat leak to the nuclear stage of a few nW, we swept the magnetic field at a rate close to that needed to maintain a constant temperature. Thus, we could achieve a linear temperature ramp while warming or cooling. The temperature was monitored with a melting curve thermometer (temperature

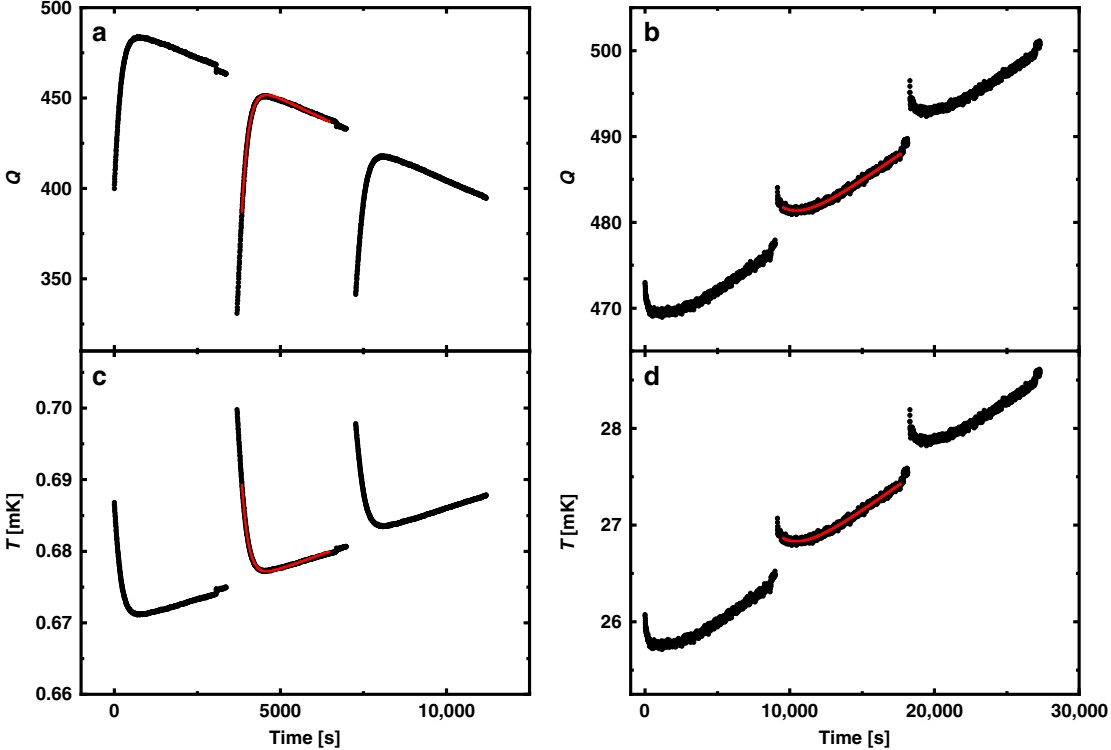

**Fig. 2 Heat pulses in normal and superfluid states. a** Typical pulses applied in the superfluid state recording the $Q$ of the IC fork vs time and **b** in the normal state at 22 bar. Also shown in red are the fitted decays to an exponential with additional linear term to account for the steady temperature drift. The $Q$ response is reversed in the normal and superfluid states, because the viscosity decreases with increased temperature above $T_c$, while it decreases below $T_c$. **c, d** The corresponding inferred temperature responses.

designated $T_{MCT}$[64] anchored to the nuclear stage. In practice, the temperature ramp of 15–35 $\mu$K·h$^{-1}$ was slow enough to allow temperature sweeps from 0.3 $T_c$ (the lowest temperature achievable in the liquid with our nuclear stage) to above $T_c$ in ~2 days (or the reverse). This time was sufficient to apply ~50 pulses in a warm-up or cooldown permitting ample time for thermal recovery between pulses. Since the thermal time constants were longer in the vicinity of $T_c$, measurements in this region were carried out with slower temperature ramps. The magnetic field on the nuclear stage was used to vary the temperature (imposing a linear magnetic field ramp) usually while warming, even up to 100 mK in the normal state.

We measured the resonant frequency, $f$, and the quality factor, $Q$, of both quartz tuning forks as a function of temperature. The $Q$ of the HEC fork was calibrated against $T_{MCT}$ during slow temperature sweeps at each pressure in the absence of pulses. Since the IC and HEC forks were nearly identical, they display similar characteristics and the $Q$ vs $T$ calibration was transferred from the HEC to the IC fork after correction for the difference in $Q^{-1}$ at $T_c$. Both quartz forks were operated in digital feedback loops (see "Methods" section to follow) and maintained near their resonant frequencies. Typical responses to applied heat pulses to the IC fork are shown in Fig. 2a, b both below and above $T_c$ at 22 bar. The signatures from the forks of thermal relaxation responses above and below the superfluid transition are inverted due to the opposite temperature dependence of the viscosity in the normal and superfluid states. The transient following a heat pulse was fit to an exponential recovery along with a linear term to account for the temperature drift. Temperature excursions from ambient were limited to a few percent and are illustrated in Fig. 2c, d.

**Normal-state measurements**. Figure 3a, b shows the recovery time, $\tau$ and the extracted thermal resistance, $R_{th}$ in Fig. 3c, d (see "Methods" section). Measurements are shown at two pressures, 0 and 22 bar. At any temperature, the inelastic mean free path at low pressure is approximately three times longer than at the higher pressure. This should allow a study of the systematics of the crossover from bulk thermal conductivity to boundary limited scattering. However, at high temperatures (>20 mK) a parallel conduction path into the IC chamber through the coin-silver walls dominates, complicating the crossover. Nevertheless, below ~10 mK, the situation is simplified since transport through the channel dominates. Two sets of calculated curves are shown: dotted lines calculated by modeling the fluid in the channel as a bulk liquid, and solid lines representing added isotropic scatterers in the channel with a density sufficient to yield a mean free path of 1.1 $\mu$m. In both cases, a parallel conduction path was added to model the heat transport via the cell walls (see "Methods" section). Vertical dashed lines mark the temperature of the superfluid transition at each pressure.

In Fig. 4, we display the effective thermal conductivity, $\kappa_{EFF} = lR_{th}^{-1}A^{-1}$ ($l$, the channel length = 100 $\mu$m, $A$, the channel cross sectional area = 3 mm × 1.1 $\mu$m) <10 mK, calculated from the measured thermal resistance and geometrical parameters of the channel. Below 10 mK, the conduction for the parallel thermal path (Kapitza resistance) is negligible and so the results represent the diffusive thermal conductivity of normal $^3$He in the channel. For both pressures, the thermal conductivity approaches a constant value. The behaviors expected for the bulk liquid ($\kappa_{EFF} \propto T^{-1}$) and for isotropic scatterers ($\kappa_{EFF} \propto T$) with a mean free path of 1.1 $\mu$m are shown as dotted and solid lines. The mean free path at 22 bar is significantly shorter at $T_c$ than at 0 bar, so the low temperature behavior is not fully developed.

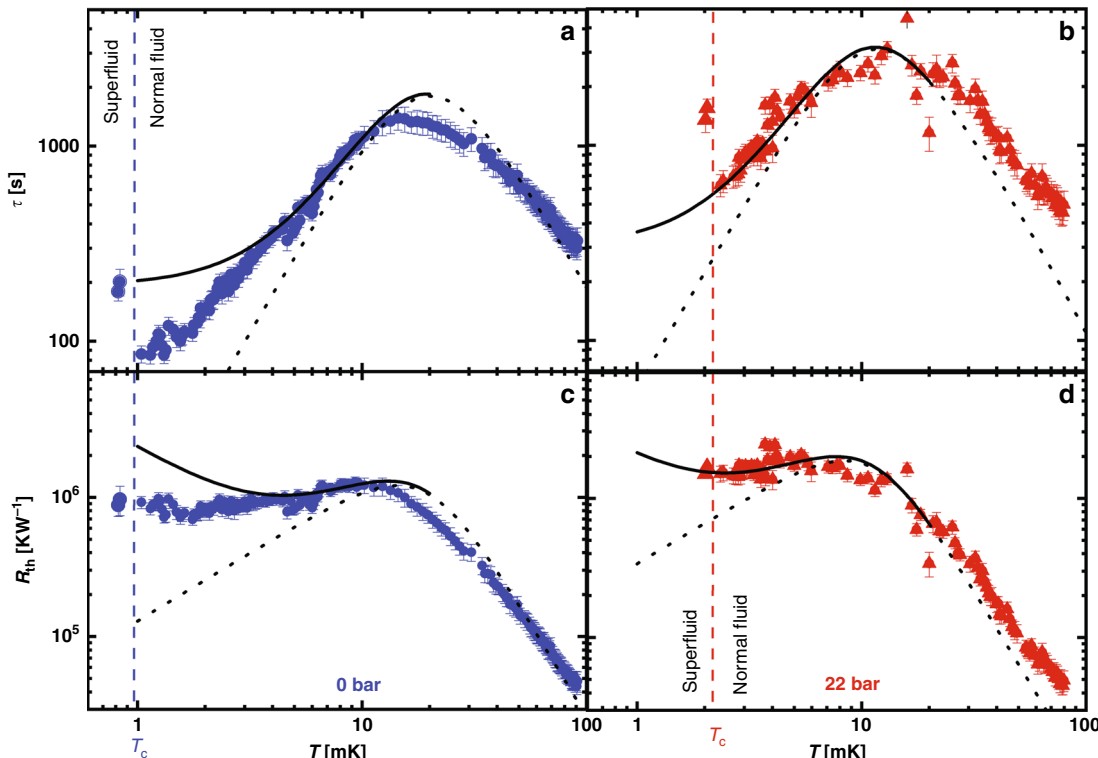

**Fig. 3 Normal-state thermal relaxation time and resistance.** The measured thermal relaxation times **a** and **b**, and calculated thermal resistances **c** and **d** at 0 (blue circles) and 22 bar (red triangles) in the normal state, showing a crossover from boundary limited behavior at high temperature to a low temperature behavior that is different from that expected for bulk. The error bars represent the standard deviation of the fits to $\tau$. Also shown is the calculated bulk behavior: black dotted lines for the bulk fluid thermal resistance in the channel in parallel with thermal boundary resistance. The black solid lines show behavior expected for an isotropic distribution of point scatterers in the channel that would give rise to a limiting mean free path of 1.1 μm. $T_c$ is marked by vertical dashed lines in blue (0 bar) and red (22 bar). The bulk calculations reference measured specific heat[63], thermal conductivity[34], and thermal boundary resistance[61].

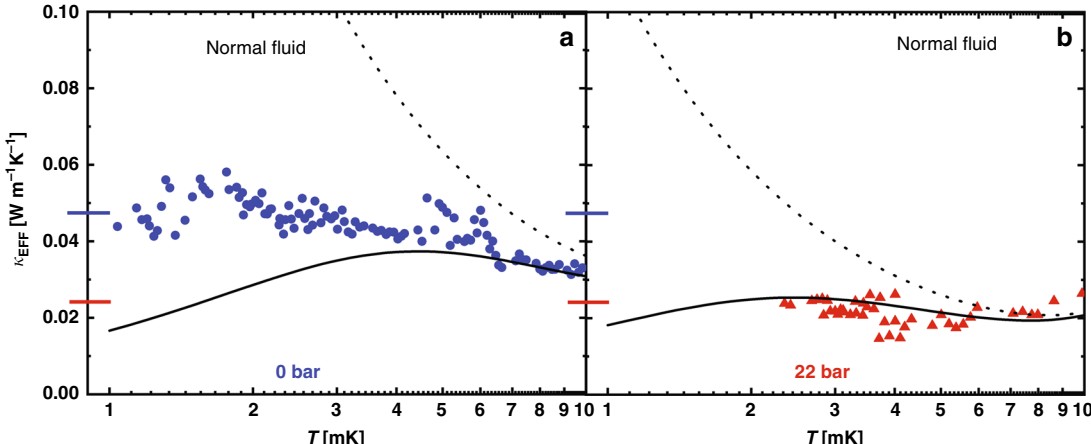

**Fig. 4 Normal state thermal conductivity.** The thermal conductivity $\kappa_{EFF}(T)$ at 0 bar (**a**—blue circles) and 22 bar (**b**—red triangles) <10 mK in the normal state calculated from geometrical parameters and $R_{th}(T)$. Also shown are the expected $T^{-1}$ bulk behaviors (dotted lines) exhibiting the $T^{-2}$ dependence of inelastic scattering of Bogoliubov quasiparticles. The solid lines show the thermal conductivity expected for a distribution of point scatterers that give rise to a mean free path of 1.1 μm. The horizontal lines define the values for $\kappa_{EFF}(T)$ at 0 bar (blue 0.047 ± 0.005 WK$^{-1}$m$^{-1}$), and 22 bar (red 0.024 ± 0.003 WK$^{-1}$m$^{-1}$) as $T_c$ is approached in the normal state.

We extrapolate $\kappa_{EFF}(T) \to$ constant as $T \to T_c$ in the normal state (where $\kappa(T_c)$ is determined to be 0.047 ± 0.005 WK$^{-1}$m$^{-1}$ (0 bar), 0.024 ± 0.003 WK$^{-1}$m$^{-1}$ (22 bar)).

At first sight, this result appears anomalous and unexpected. The thermal conductivity, $\kappa$ of a Fermi liquid is proportional to the thermal mean free path $\lambda_\kappa$ (see "Introduction" section). In a bulk Fermi liquid in the absence of impurities, $\lambda_\kappa \propto T^{-2}$ yielding

$\kappa \propto T^{-1}$ since $C_v \propto T$. Impurities lead to a temperature independent contribution to the mean free path[36] and hence $\kappa \propto T$, and this might also be expected for boundary limited scattering (also see "Methods" section). The onset temperature for such boundary limited scattering can be estimated from the bulk thermal mean free path $\lambda_\kappa T^2 = 23.6$ μm·mK$^2$ at 0 bar and 7.36 μm·mK$^2$ at 22 bar[34]. At the lowest pressure, we expect strong mean free path

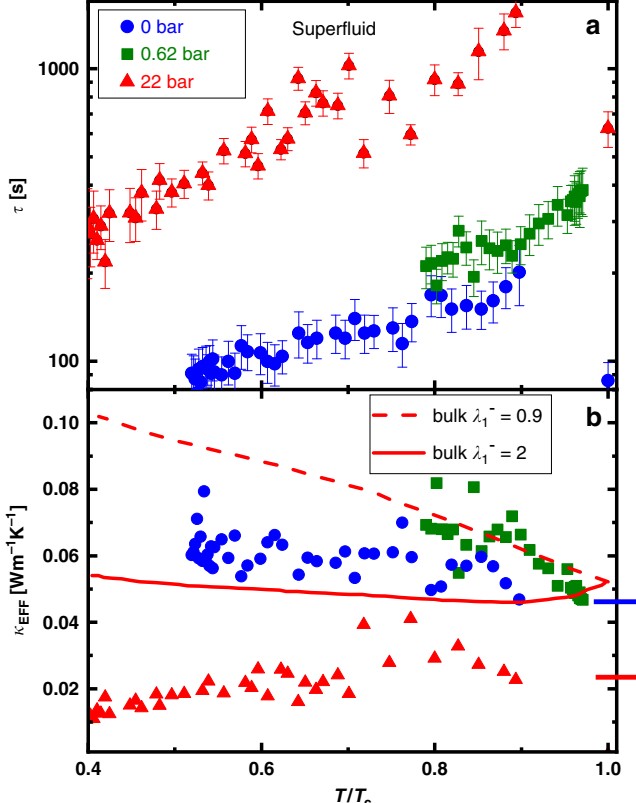

**Fig. 5 Superfluid state relaxation times and thermal conductivity.** The measured thermal relaxation times (**a**) and calculated effective thermal conductivities, $\kappa_{\mathrm{EFF}}$ (**b**) at 0 (blue circles), 0.62 bar (green squares), and 22 bar (red triangles) below the superfluid transition temperature $T_c$. The error bars represent the standard deviations of the fits to $\tau$. The bulk calculations reference calculated thermal conductivity[47] at 21 bar with $\lambda_1^- = 2$ (red solid line), $= 0.9$ (red dashed line), and should be compared to the 22 bar data. The blue (0 bar) and red (22 bar) horizontal lines mark the limiting ($T \rightarrow T_c$) value of $\kappa_{\mathrm{EFF}}$ in the normal state (see Fig. 4), and are well below the bulk-like conductivity through this channel. In the bulk and under confinement, $\kappa_{\mathrm{EFF}}$ is only weakly temperature dependent below $T_c$. No calculations of $\kappa(T/T_c)$ exist in the literature for 0 bar.

effects in the 1.1 μm channel below ~5 mK and at 22 bar below ~2.6 mK (near $T_c$).

As shown in Figs. 3c and 4a, the observed behavior for the thermal conductivity in our restricted flow channel at 0 bar (blue circles) clearly lies between the bulk and impurity limit values. At 22 bar (Fig. 4b red triangles), the data lie well below the expected bulk behavior. The observed thermal conductivity at both pressures <5 mK is temperature independent, suggesting an effective mean free path proportional to $T^{-1}$. (In the case of 0 bar, the mean free path increases by a factor of >25 below 5 mK before $T_c$ is attained, and the measured thermal resistance can be distinguished from the impurity model's $T$ dependence. At 22 bar, the mean free path is insufficiently long; thus, the results cannot be distinguished from the impurity limit behavior before $T_c$ is attained). While inconsistent with Fermi liquid theory, such an unusual temperature dependence of the mean free path has previously been inferred in studies of mass transport in ³He films over polished silver surfaces[44], which found a momentum current relaxation time $(\tau_\eta) \propto T^{-1}$ at temperatures <100 mK. This result was interpreted[43] in terms of quasiclassical interference between bulk and boundary scattering channels, as earlier proposed for thin metal films with rough surfaces[45,46,65]. This theory has subsequently been extended to thermal transport[66], yielding a

constant value of $\kappa$ over a wide temperature range for surface roughness of 3 nm r.m.s. with fractal correlations. This is likely to be beyond the upper limit for roughness of our glass substrate[62,66]. A more likely source of scattering is the presence of trapped charges at the surface of the silicon[67]. Such charges (due to dangling bonds) would induce local density variations that would create random scattering potentials, mimicking surface roughness. Future studies where the silicon surface is passivated[68] should reveal whether this hypothesis is tenable.

**Superfluid state.** Below the superfluid transition temperature $T_c$, we carried out experiments at 0, 0.62, and 22 bar. The lowest pressure (0 bar) was chosen because the inelastic mean free path in both the normal and superfluid states would be the longest. At the nearby pressure of 0.62 bar, $T_c$ is almost 10% above its value at 0 bar and the inelastic mean free path at $T_c$ is already ~20% shorter than at 0 bar. At 22 bar, the inelastic mean free path is much smaller than the other two pressures and there is only a small temperature window, in which the A phase will be present in the two bulk chambers near $T_c$. (Two superfluid phases are found in bulk ³He in zero magnetic field: the anisotropic A phase occupies the high pressure $P \geq 21.22$ bar region near $T_c$, the isotropic B-phase occupies the remainder of $P$, $T$ space[64]). Confinement of superfluid ³He in the channel of height $d = 1.1$ μm modifies this phase diagram[8,69] and stabilizes the A phase over a significant temperature range at low pressures (to 0.7 $T_c$ (0 bar), 0.77 $T_c$ (0.62 bar), and 0.93 $T_c$ (22 bar)).

The measured relaxation times, $\tau$, in the superfluid state are plotted in Fig. 5a and the inferred thermal conductivity $\kappa_{\mathrm{EFF}}(T)$ in the channel is shown in Fig. 5b. For comparison, we show the results of model calculations (for bulk superfluid ³He-B) from Einzel[47] at 21 bar. The precision of the experiment did not reveal differences in $\tau$ (and thus the thermal resistance) through the expected A–B transitions in the channel. Our results in the superfluid state (Fig. 5b) constitute a measurement of the thermal conductivity under confinement in the absence of hydrodynamic heat flow. A strong contribution from thermal counterflow[53,54] has been observed in bulk superfluid ³He. In contrast, no such enhanced thermal conduction was observed in this experiment below $T_c$. Under confinement, the normal fluid is expected to be clamped and so the thermal counterflow contribution should be limited. The hydrodynamic thermal conduction is first estimated using extrapolations of the measured effective viscosity[55] under relatively modest confinement and found to be small (Supplementary Fig. 1). This predicts (in Supplementary Note 1) that comparable diffusive and hydrodynamic contributions to thermal conduction would arise if $d$, the confinement height were of the order of 100 μm–1 mm, due to the $d^{-3}$ contribution to the impedance $Z$.

There has been no prior systematic experimental study of the diffusive thermal conductivity of superfluid ³He. However, the theory of spin-independent transport has been developed for bulk[47,70,71], and is parameterized by a series of well-defined relaxation times and a pressure-dependent scattering parameter, which for the thermal conductivity is referred to as $\lambda_1^-$. Over a reasonable range of possible values for $\lambda_1^-$ ($0.9 \leq \lambda_1^- \leq 2.0$), the thermal conductivity in the B-phase at 21 bar at low temperatures, ($T/T_c = 0.4$), relative to that at $T_c$ varies by at most a factor of two[47]. We have seen that at both low pressure and 22 bar, under strong confinement, $\kappa_{\mathrm{EFF}}(T_c)$ is reduced below the bulk value at variance with standard theory of the normal state. In Fig. 5b, we compare our measurements at 22 bar with the theory for bulk diffusive thermal conductivity at 21 bar. The thermal conductivity at 22 bar shows a weak maximum and then decreases as the temperature is lowered below $T_c$. At low pressure

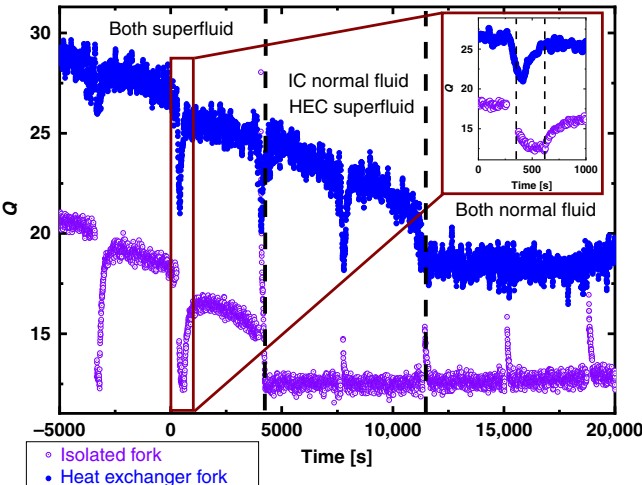

**Fig. 6 Quality factor ($Q$) of both forks near $T_c$.** The $Q$ as a function of time for the two forks (purple IC, blue HEC) near $T_c$ taken at 0 bar. Pulses applied were separated by 60 min and the nuclear stage was warmed up at a constant rate around 10 $\mu$K·h$^{-1}$. When both the IC and HEC are in the superfluid state (first three pulses) a strong, and unexpected, response is seen in $Q_{HEC}$. As the IC passes through its $T_c$, the third pulse applied evokes a response in $Q_{HEC}$. Even when the IC is in the normal state (the response in $Q$ of pulse 4 is reversed), a strong response is seen in $Q_{HEC}$. Only when both the HEC and IC are in the normal state (pulses 5–7) is there no anomalous response seen in $Q_{HEC}$ to a heat pulse in the IC. The inset shows details of the second pulse. The first dashed line marks the end of the pulse applied to the IC and the second dashed line marks where the IC, which was heated into the normal state by the pulse, passes through $T_c$. The HEC fork response is maximum at the end of the pulse applied to the IC.

the thermal conductivity increases before plateauing at approximately twice its value at $T_c$, similar to the predicted behavior that the expected conductivity should increase or remain nearly constant, depending on the choice of scattering parameter. Of course, the value of $\kappa_{EFF}$ at $T_c$ is significantly reduced below the bulk value. Thus, a full calculation of the diffusive thermal conductivity under different degrees of confinement is highly desirable. For channels with intermediate confinement of order the mean free path, this would require extension of the results from mass transport[55] to thermal transport, and include a treatment of surface slip and surface Andreev scattering into the extreme Knudsen regime. Under conditions of strong confinement for which the channel height is comparable to the superfluid coherence length (typically <1 $\mu$m at zero bar), a full quasiclassical calculation incorporating the contributions of surface bound states is required.

**Anomalous response.** We now report on the anomalous thermal responses detected by the HEC fork. In this series of experiments, we applied long duration pulses (between 100 and 300 s to deposit more heat) to the IC fork resulting in a change in temperature to the IC of order $\Delta T_{IC} \sim 5$–10%. Surprisingly, we observed a small but immediate response in the HEC fork, decaying with a time constant similar to that for thermal relaxation in the IC. The response of the HEC fork in the vicinity of $T_c$ at 0 bar is shown in Fig. 6. This shows that the response is associated with superfluidity in the channel, since it vanishes in the normal state, which also rules out electrical crosstalk. The response is essentially immediate, and correlates with the temperature rise of the IC (see inset to Fig. 6). It cannot correspond to heating of the HEC due to thermal relaxation between the two chambers. The magnitude of the apparent temperature rise is too

large and too fast, given the total heat deposited by the IC fork and the relatively large heat capacity of the $^3$He in the HEC. The hydrodynamic heat flux is only a small fraction of the observed heat conduction and would be too small to measure (Supplementary Note 1). Neither can it arise from a transient increase in temperature in the HEC arising from superflow from HEC to IC. In Supplementary Note 2, we show that any superfluid "deficit" (arising from superflow from the HEC to the IC) is too small to lead to observable temperature changes.

The characteristics of the response thus show that the HEC fork is not registering a global change in the temperature of the HEC. The registered response in the HEC fork must therefore be local and we denote it as $T^{HEC*}$. We propose that the anomalous and unexpected response is attributable to a flux of excitations incident on the HEC fork, driven by the fountain pressure generated in the IC.

Before further discussion, we first report the systematics of this effect for the different reduced temperatures and pressures investigated. In Fig. 7a–i, we show plots of the local temperature of the HEC fork $T^{HEC*}/T_c$ at 0 bar (Fig. 7a–c), 0.62 bar (Fig. 7d–f), and 22 bar (Fig. 7g–i), compared to the temperature pulse in the IC for a selection of reduced temperatures. They show that the strongest anomalous response is seen at the lowest temperatures, and is significantly weaker at the highest pressure. At 22 bar, no response in the HEC fork is seen for $T/T_c \geq 0.6$, while at 0 bar the response persists to $T_c$ (Figs. 6 and 7a–c). Complete traces across the full temperature range in the superfluid of the observed change in $T^{HEC*}$, $T^{IC}$ after the application of pulses to the IC fork are shown for the three pressures measured in Supplementary Fig. 2. We also show (in Supplementary Fig. 3) the evolution of the $Q$ in both HEC and IC forks, following pulses applied to the HEC fork, with comparable temperature excursions to those previously created using the IC fork. These results show that the anomalous heat flow is bidirectional. The signals on both forks are comparable in this case; it is likely that the location of the IC fork at the throat of the channel (Fig. 1b) enables the efficient detection of the signal. There are also indications that the flow of excitations may be subject to limitation above some threshold flux (Supplementary Note 3 and Supplementary Fig. 4).

We now comment on how the detection of these transient responses is correlated with the viscous mean free path, motivated by the fact that a mean free path long compared to the channel height (Knudsen regime) promotes the slip of normal fluid in response to the fountain pressure. A consistent picture emerges from the three pressures studied. In Fig. 8, we show a plot of the viscous mean free path[55] as a function of reduced temperature $T/T_c$. For 0 bar at $T_c$, the viscous mean free path, $\lambda_\eta$, is ~72 $\mu$m. It then decreases rapidly by about a factor of 2 below $T_c$ before rising exponentially at low temperatures, ensuring that the channel is well within the Knudsen regime at all temperatures below $T_c$. At the intermediate pressure (0.62 bar), the received signal in the HEC fork is present at $T_c$, then following trends in the mean free path, it gets weaker below $T_c$ before growing as the temperature is further lowered. This temperature dependence can be seen in Fig. 7d–f and in the continuous trace data (and inset) shown in Supplementary Fig. 2. The smallest response at 0.62 bar is aligned with the location of the minimum in the viscous mean free path (Fig. 8). At the highest pressure, 22 bar, the mean free path at all reduced temperatures is significantly smaller that at low pressure. Consistent with this, the anomalous response is only observed at $T/T_c \leq 0.6$, where the mean free path is sufficiently large ($\lambda_\eta \geq$ 6 $\mu$m, Fig. 8). In conclusion, the observations support the hypothesis that the anomalous heat conduction mechanism appears in the strong Knudsen regime.

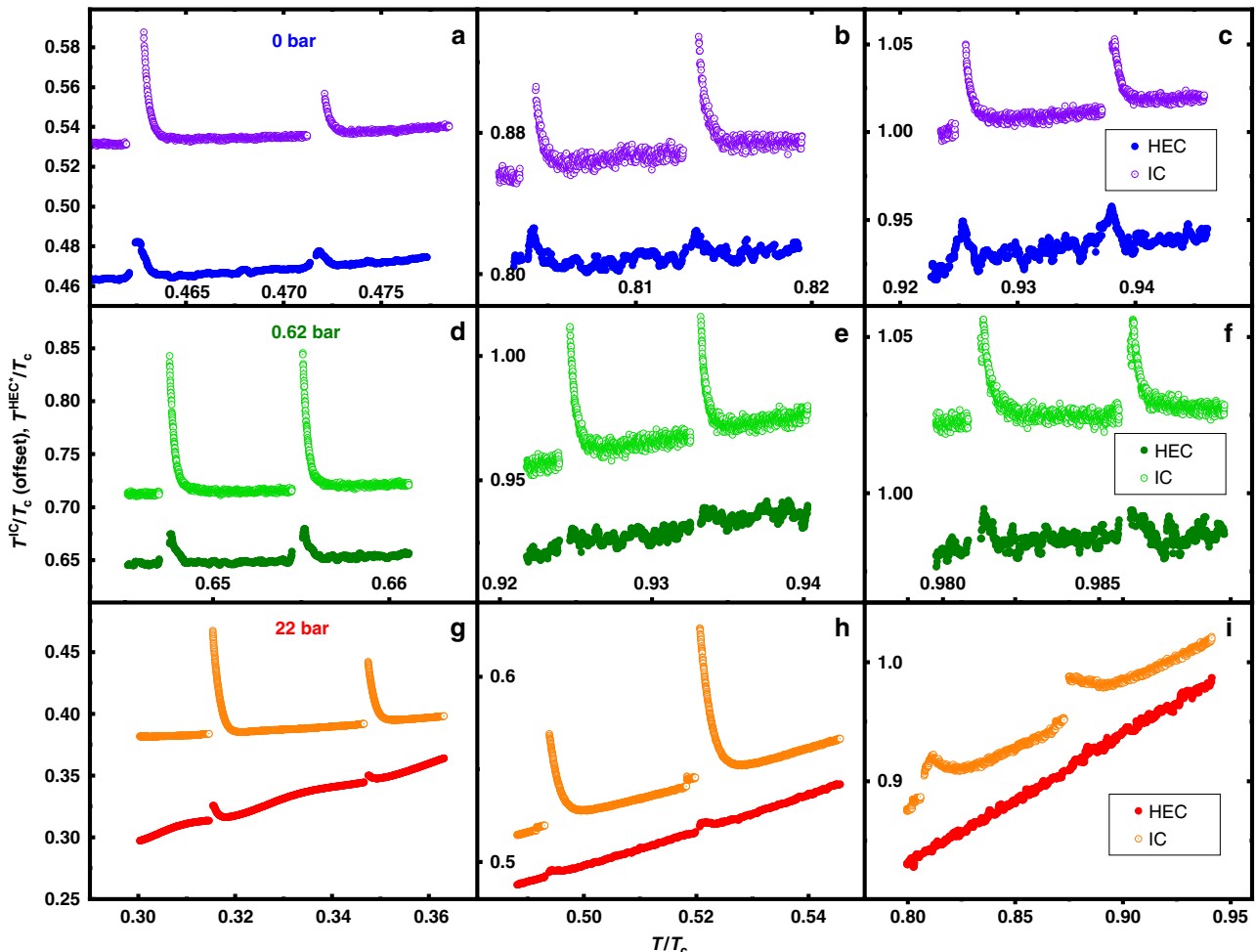

**Fig. 7 Local temperatures of both forks after pulses at representative temperatures.** The 3 × 3 panels (**a–i**) show the local temperature $T^{HEC^*}/T_c$ (filled circles) compared to $T^{IC}/T_c$ (open circles) (offset upward by 0.05 $T^{IC}/T_c$ for clarity) against $T/T_c$, the slowly rising temperature of the nuclear demagnetization stage. The top row shows three representative sets of pulses at 0 bar and near 0.45 $T/T_c$ (**a**), 0.8 $T/T_c$ (**b**), and 0.94 $T/T_c$ (**c**). The middle row (**d–f**) shows results for 0.62 bar, and the lower row shows results at 22 bar (**g–i**). Two different pulse durations were used accounting for differences in the initial temperature rise observed. Data obtained while cooling exhibits the same behavior.

## Discussion

The unanticipated discovery of the anomalous responses in the superfluid reported in the previous section poses a puzzle. We observe a thermal response that is clearly a local heating effect. It propagates through the narrow channel between forks whose separation is very much greater than the bulk quasiparticle mean free path. Thus, neither conventional hydrodynamic heat flow, nor ballistic propagation of bulk quasiparticles can be responsible for this effect or account for its bidirectionality. We therefore consider the potential role of surface excitations in our highly confined channel ($d/\xi_0 \sim 15$ at $P = 0$ bar, $d$ being channel height, $\xi_0$, the coherence length). We stress that dynamics in this long mean free path and highly restricted geometry regime have not been explored previously theoretically or experimentally in superfluid $^3$He. At these large Knudsen numbers, in an analysis of hydrodynamic flow, the normal fluid velocity must be constant across the height of the channel (plug flow). The flow velocity will thus be affected by Andreev scattering processes at the surface; the silicon surfaces are close to atomically flat. This corresponds to a flux of quasiparticles propagating in the 1.1 μm channel. However, the separation between the HEC fork and the opening of this channel is several mm, much longer than the quasiparticle mean free path. It therefore does not seem possible that a significant flux of bulk quasiparticles can transit through the intervening superfluid.

Under such strong confinement, the influence of the surfaces on both the order parameter and surface bound excitations must play a dominant role, extending across the entire channel[56–59]. Both the order parameter and the density of surface excitations can be calculated self-consistently, using quasiclassical theory. Currently, the dynamics of these surface excitations[11] is not fully understood. Furthermore, the interplay with bulk quasiparticle excitations under none-quilibrium conditions is also of interest. For example, there have been recent studies on mechanical resonators immersed in superfluid $^3$He, in which these effects seem to play an important role[60,72,73].

These considerations lead us to the following conjecture. Let us assume a flux of surface excitations, driven by the fountain pressure between the two chambers and with an accompanying counterflow of superfluid, which also Doppler shifts the energy of the surface excitations[60,73]. If we can identify a mechanism by which surface excitations could be injected into the bulk, and if the interaction between surface excitations and bulk excitations is weak (as suggested in ref. [60] who measured lifetimes of order 6 ms), they may move with little attenuation over distances much longer than the inelastic mean free path for bulk quasiparticles, and so account for the local transient response, $\Delta T^{HEC^*}$ seen by the HEC fork.

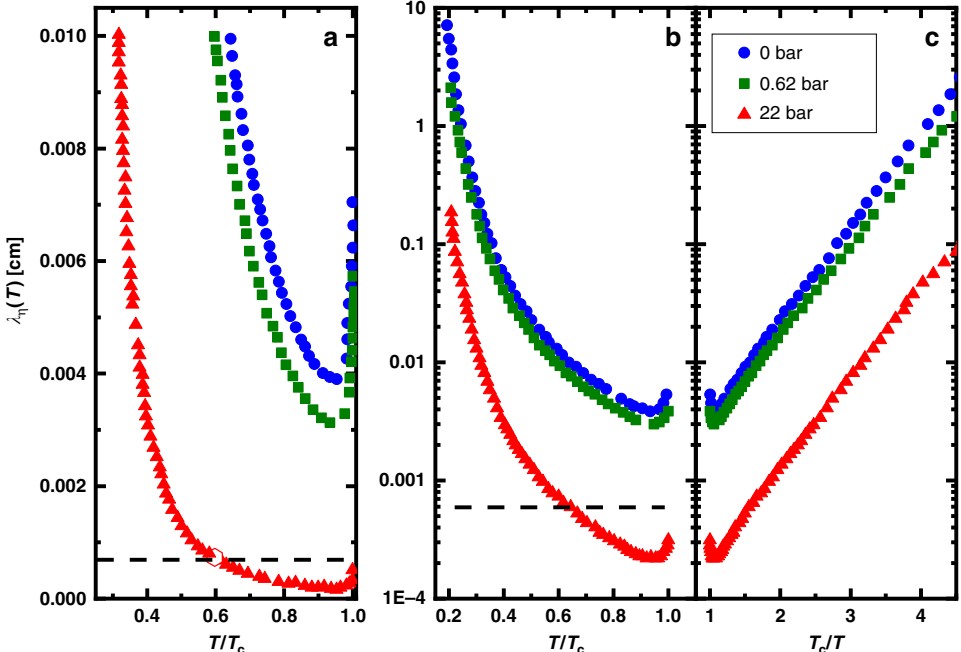

**Fig. 8 Viscous mean free path in superfluid state. a** The viscous mean free path, $\lambda_\eta$ below $T_c$ as a function of $T/T_c$ and $T_c/T$ for 0 (blue circles), 0.62 (green squares), and 22 (red triangles) bar pressures. Note the strong increase in $\lambda_\eta$ from a minimum just below $T_c$. **b, c** The viscous mean free path in cm on a logarithmic scale. At 0.6 $T/T_c$ for 22 bar, the mean free path is ~6 µm (dashed line).

The sharp corners located at the throat of the channel may play a role in the release of surface excitations into bulk. This could arise from the combination a large local superfluid velocity at the corner, arising from potential flow of superfluid, and the superfluid gap profile in the vicinity of the corner. At this stage, a quantitative model is beyond the scope of this experimental report.

Surface excitations are sensitive to the surface boundary layer of helium. In the present case, this is a magnetic surface layer of $^3$He, which is found to increase the density of low-energy excitations relative to a nonmagnetic diffusely scattering surface[6]. Quasiparticle scattering at the surface can be systematically controlled by adjusting the surface boundary layer by $^4$He plating, since $^4$He preferentially binds to surfaces. Further, thermal transport measurements under various surface scattering conditions are clearly desirable.

As reported, the experiments so far have not resolved any systematic influence of the particularities of the superfluid phase stabilized in the strongly confined channel. At 0 bar the A-phase should be stable in the channel[69] down to $T/T_c = 0.7$ (0.93 at 22 bar), with an A–B interface at each end between the 1.1 µm channel and the lead-in channel, where the bulk B-phase is expected. Furthermore there is experimental evidence of a two-dimensional spatially modulated phase in a channel of comparable height at zero pressure[8]. The stable phase within the channel will influence the nature of the surface excitations. For nonmagnetic specular surfaces a domain wall in the planar distorted B-phases depletes the density of mid-gap low-energy surface bound states[74]. Thus, while we expect that there will be a rich interplay of superfluid phase, domain walls, and superfluid interfaces on the thermal transport under strong confinement, this is beyond the scope of the current experiment, and will be the subject of future investigation.

In conclusion, we have made a study of thermal transport through a 1.1 µm tall cavity in both the normal and superfluid phases of $^3$He. There are three principal findings.

First, the effective thermal conductivity of normal $^3$He under this strong confinement is essentially temperature independent <10 mK at low pressure and <5 mK at 22 bar. Consequently the magnitude of the conductivity at $T_c$ is quantitatively different from that in bulk. The temperature independence can be understood in terms of an effective thermal mean free path that varies as $T^{-1}$, rather than $T^{-2}$ (bulk inelastic mean free path) or constant (boundary limited scattering). This is qualitatively consistent with previous studies of mass transport in thin films, that is accounted for by a theory of interference between inelastic scattering within the film and elastic scattering, arising from an effective disorder potential originating at the surface.

Second, the relatively weak temperature dependence of the thermal conductivity under strong confinement measured in the superfluid state, relative to its value at $T_c$, is similar to that calculated for bulk liquid, and reaffirms the absence of dominance by hydrodynamic transport. This result motivates further measurements of thermal transport in a slab-like cavities, sufficiently confined to make hydrodynamic heat flow small compared with diffusive heat flow, but as large as possible to minimize the effects of surface slip, and minimize the contribution of surface states. By contrast, the height of the present cavity was chosen to approach the superfluid coherence length at the lowest pressures. In this case, a full quasiclassical calculation incorporating the contribution of surface excitations to the diffusive thermal transport is highly desirable.

Third, despite the fact that according to our estimates, the overall thermal transport should be dominated by diffusive thermal transport, we observe an anomalous thermal response that appears to be driven by the fountain pressure difference between the two chambers (in either direction) provided the Bogoliubov quasiparticle mean free path is long enough (apparently ≥6 µm). This appears to correspond to a local nonequilibrium response, which we suggest is evidence of quasiballistic thermal transport due to unbound long-lived surface excitations that can propagate through the bulk.

With attainable improvements in the precision of thermometry, this work opens the prospect of a variety of thermal transport studies of topological superfluid $^3$He under strong confinement at length scales comparable to the superfluid coherence length. Thermal transport should be sensitive to the presence of interfaces in the superfluid, either those arising spontaneously as in the spatially modulated superfluid, or those engineered by steps in cavity height, since confinement controls the stable superfluid order parameter. As recently emphasized theoretically[19], the detection of surface, edge, and interface excitations by thermal transport, and the thermal Hall effect and edge currents in topological superfluid $^3$He should act as a benchmark for similar studies of putative topological superconductors.

## Methods

**Thermal conduction channel construction.** We fabricated the entire assembly (IC, thermal conduction channel holder, HEC) out of coin-silver (Fig. 1 and Supplementary Fig. 5). This minimised time-dependent heat leaks into the $^3$He because the walls were thermally well anchored to the nuclear stage. The 5 mm × 5 mm silicon chip (Fig. 1a) that comprises one face of the nanofabricated channel, and establishes the thermal impedance between the two chambers for $^3$He was made at Cornell's nanofabrication facility using our well-established process flow[62]. After patterning of the silicon, a matching square piece of highly polished sodium-doped glass (Hoya SD-2) was bonded to the silicon. After bonding, the edges of the silicon and glass that were parallel to the heat flow were rounded off, using a high-speed Dremel tool and carborundum bit. The cavity was mounted (using Trabond 2151 epoxy) into a coin-silver holder that had walls that were machined to a 0.15 mm thickness (Supplementary Fig. 5b). The rounded corners distribute stress on the silicon and glass components during thermal cycling. The thin coin-silver of the holder also enabled movement of the metal with the epoxy and silicon-glass cavity, so as to accommodate thermal contraction on cooling. A dummy cavity (without a through pathway) was cycled repeatedly to liquid nitrogen temperatures and proved to be leak tight post-cycling. The design is such that there should be no differential pressure across the large faces of the cavity, thus no additional pressure-dependent bowing should be present to alter the cavity dimensions as the pressure is varied[69].

**Fork operation and pulses.** The forks were driven using a constant voltage signal at a level small enough so that no drive-dependent heating was observed. The detection was via a voltage preamplifier connected to one tine of the fork, while the drive was applied to the adjacent tine. We estimate from the energy deposited during heat pulses and the ratio of drive voltages that the ambient heating due to operation of the forks is ~0.1 pW near 1 mK. The preamplifiers for each fork had their 6 dB/octave filters set at 10 and 100 kHz. In order for the feedback loop operate well at low temperatures (where the $Q$ is low, approaching 10 at $T_c$ at 0 bar for a resonant frequency ~34 kHz), we measured (at 20 mK) and fitted the background (nonresonant signal) over a wide frequency range (10–60 kHz) using a fifth-order polynomial after excluding the region of the resonance. After subtraction of the nonresonant background signal, the inferred $Q$ was reliably indicative of the temperature of the liquid. Further sweeps were also carried out at lower temperatures <3 mK where the $Q$ was lower to obtain better fits, and identify and compensate for a small temperature-dependent background. After the background was well fit, we could carry out a calibrating sweep at intermediate temperature (typically 10 mK) where the $Q$ was ~200, to establish the conversion from peak amplitude to $Q$ and then measure the temperature-dependent real and imaginary components of the response, while driving the forks at a constant frequency. If the entrained mass caused a shift in the resonant frequency that exceeded 10% of the linewidth, we recomputed the resonant frequency and altered the drive frequency to coincide with the center frequency. Nearly smooth responses of the real and imaginary components of the recorded signal across these rebalances assured us that the fits were accurate. The technique allowed us to track the $Q$ of the forks across $T_c$ at 0 bar, where the viscosity is largest.

Pulses were applied by increasing the drive voltage above ambient by a factor of 10. We could apply more heat as needed by increasing the duration of the high drive. During the high-drive state, it was not possible to track the resonant frequency of the fork or its $Q$. Therefore, we turned off the frequency rebalance component of the program and operated at a fixed drive frequency during the pulse. However, we forced a rebalance of the forks prior to applying the pulse, so that they would both be operating close to their individual resonant frequency during the recovery after the pulse. The rebalance is visible as a small discontinuity prior to the application of the pulse in the inferred $Q$ vs time shown in Fig. 2. In practice, we found the $Q$ to be more robust against any background corrections (compared to the inferred resonance frequency) so the temperature was monitored using the $Q$.

**Calculation of thermal resistance.** The measured $\tau$ values in the normal state were converted to an effective thermal resistance by evaluating the heat capacity of the IC using interpolations[63]. We used the relationship $\tau = R_{th}C$, where $R_{th}$ and $C$ are the effective thermal resistance to the IC and heat capacity of the $^3$He in the IC, respectively. The derived values of $R_{th}$ are subject to a further systematic 10% error accruing from the estimate of the volume in the IC and thus $C$. The thermal conductivity of $^3$He is relatively poor at high temperatures because the excitation density in this regime leads to a short mean free path[75]. In our arrangement, a parallel path for heat transport becomes significant above ~10 mK, through the Kapitza boundary resistance, $R_K$, of the ~14 cm$^2$ area[61]. Generally at low temperatures, the boundary resistance of a sintered heat exchanger varies as $T^{-3}$, and below ~15 mK, there is an abrupt change in power law to a $T^{-1}$ behavior[76]. The origin of this crossover is poorly understood and may be due to the localization of phonons in the sinter[77]. Since the surface of the IC is not composed of sinter, but instead is as-machined metal, we use the results from ref. [61] ($R_K = 0.08\,T^{-2.685}A^{-1}$ (K m$^2$W$^{-1}$), with $A$ the area in m$^2$, $T$ the temperature in K) after scaling for the difference in sound velocity and density of the dilute mixture as compared to pure $^3$He at 0 bar. In addition, to account for changes in density and sound velocity with pressure, we have to further scale[78] the Kapitza resistance by the ratio of molar volume at pressure $P$ to the molar volume at pressure $P = 0$ ($V_m(P)/V_m(0)$), and the ratio of sound velocity ($c_1$) at $P = 0$ to the sound velocity at pressure $P$ ($c_1(0)/c_1(P)$). The calculated values of the thermal relaxation time show a crossover at ~10–20 mK from surface dominated behavior at high temperatures to the channel dominated resistance. The actual behavior (Fig. 3c, d), especially at 22 bar, does not follow the $T^{-3}$ power law, for two reasons. First, the measured Kapitza resistance for a sheet obeys a power law that is closer to $T^{-2.7}$ for this temperature range[61]. Second, a portion of the surface area in the IC is in the form of two closely fitting cylinders. The effective area of the cylinders is modified by the conductivity of the $^3$He that fills the gap between the cylinders: the area participating in heat flow decreases as the temperature increases due to the temperature variation of the conductivity of the $^3$He.

In Figs. 3 and 4, we also include the expected behavior (black solid lines) for the thermal conductivity of samples with similar geometry to that studied whose resistance is characteristic of a uniform distribution of elastic scatterers spaced to yield a 1.1 μm elastic scattering length. We modify $\kappa = 1/3(C_v/V_m)v_F^2\tau_\kappa$ by replacing $\tau_\kappa$ with an effective scattering time, $\tau_{eff}$ given by Mathiessen's rule ($\tau_{eff}^{-1} = \tau_{el}^{-1} + \tau_{in}^{-1}$). Thus, $\tau_{eff}$ approaches a constant when the quasiparticle scattering time, $\tau_{in}$, exceeds the impurity scattering time, $\tau_{el} = 1.1$ μm $v_F^{-1}$. The resulting impurity dominated thermal conductivity thus varies as $T$.

We also calculated the corresponding values of $R_K$ using the results obtained for silver sinter[76], that show a crossover from 0.04 $T^{-3}A^{-1}$ (K m$^2$W$^{-1}$) to 250 $T^{-1}A^{-1}$ (K m$^2$W$^{-1}$) behavior <12.6 mK ($T$ is the temperature in K, $A$, the geometric area of wetted metal, in m$^2$). When we calculate the expected values of $\tau$ and $R_{th}$ after scaling for pressure, we find little difference between the effective thermal resistance <10 mK (Supplementary Figs. 6 and 7), where the results using the behavior observed in sinter[76] are shown in gray and the results using the results for a sheet[61] are shown in black. Supplementary Fig. 7 shows that in either case, the thermal conduction through the channel dominates over the Kapitza resistance <10 mK, the region of interest here.

## Data availability

The data that supports this study will be made available through Cornell University e-commons data repository at https://doi.org/10.7298/4fhq-e356.

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

## Acknowledgements

We acknowledge input from Prof. J.A. Sauls including access to preliminary calculations and thank Prof. A. Golov for providing a copy of the thesis of Wellard. We also acknowledge a helpful exchange with V. Ngampruetikorn regarding contributions of surfaces to thermal conduction. This work was supported at Cornell by the NSF under DMR-1708341, 2002692 (Parpia), PHY-1806357 (Mueller), in London by the EPSRC under EP/J022004/1. John Wilson's participation was supported in part by the Cornell Center for Materials research with funding from the Research Experience for Undergraduates program (DMR-1719875). In addition, the research leading to these results has received funding from the European Union's Horizon 2020 Research and Innovation Programme, under Grant Agreement no. 824109. Fabrication was carried out at the Cornell Nanoscale Science and Technology Facility (CNF) with assistance and advice from technical staff. The CNF is a member of the National Nanotechnology Coordinated Infrastructure (NNCI), which is supported by the National Science Foundation (Grant NNCI-1542081).

## Author contributions

Experimental work and analysis were principally carried out by D.L. with early contributions by A.E., assisted by M.T and J.W., with further support from E.N.S. and J.M.P. Presentation of figures was the joint work of A.E. and D.L. N.Z. had established most of the routines for the phase locked loop operation of the quartz fork for earlier experiments. E.M. provided general guidance on thermal conductivity issues, and significantly contributed to the data analysis protocols and the writing of the manuscript, and N.Z. and T.S.A. established and carried out the nanofabrication protocols. D.E. calculated the viscous and thermal mean free paths. J.M.P. supervised the work, and J.M.P. and J.S. had leading roles in formulating the research and writing this paper. All authors contributed to revisions to the paper.

## Competing interests

The authors declare no competing interests.
