## [Peer Review File · Nature Communications]

Reviewers' comments:

Reviewer #1 (Remarks to the Author):

This has been a challenging paper to review. On the one hand it details brand new experiments that are extremely difficult to carry out, and deserves high profile publication on that basis alone. But on the other hand some of the data, and its analysis, feels like preliminary work that anticipates much larger breakthroughs to come, making one question whether Nat Comms is the right home for publication. The experimental results will certainly be of great interest to others in the community; indeed, these are experiments on helium-3 in a completely new regime of confinement. However, the link to other quantum systems, touched on in the introductory material, is really only cursorily made, potentially alienating a more general reader. Much (commendable) effort is spent on ruling out particular interpretations of the data, but often the reader is then left with speculations of what effects may be being measured, rather than concrete demonstrations of new physics with predictions for and links to the behaviour of other quantum matter systems.

That said, the work is stimulating, and has the potential to influence others in the field, both in terms of laying down the gauntlet to theorists, and opening the door for other experimentalists to reproduce and build on what has been done here. I did not find any obvious errors or seriously incorrect statements. The experimental team are world experts who know their territory well, and also do a very good job of setting the work against a background of previous relevant research.

Should the authors be given the chance to improve the work (I think they should) then I have a number of recommendations.

1. All figures, both schematics and graphs, could be greatly improved with consistent style, labelling and keys.
2. I do not agree with the statement on Page 13 that the data in Figs 3(b) and 4 clearly lie between the extremes. The 22 bar data appear to follow the point scatterer theory lines very well below 10mK. This should be noted, commented and compared.
3. If possible I would like to see more argument made for the existence of transport due to surface excitations. In my view this is the most important and provocative result of this work. Augmenting this aspect may require a re-structure to reduce the focus on the normal state measurements.
4. Attempt to make clearer, more quantitative statements of how helium-3 in this slab geometry can be a model for other quantum matter systems, particularly superconducting ones.

Reviewer #2 (Remarks to the Author):

This work presents thermal conductivity measurements in normal and superfluid state of helium-3 in a 1.1 μm channel using a relaxation method. The main results can be summarized:

1. In normal fluid, the thermal conductivity exhibits the temperature dependence indicating that the effective thermal mean free path varies as T^{-1} . This result is qualitatively consistent with their previous work of mass transport.
2. In superfluid, they were able to investigate the dominant diffusive conduction mechanism by suppressing the hydrodynamic counterflow in the channel.
3. An unusual thermal interaction between two cells separated by the channel was also observed. The authors speculate that the unexpectedly fast thermal response in a cell when heat was generated in

the other may be evidence of quasi-ballistic thermal transport due to the surface excitations.

Overall, data are technically sound and the results are novel and important in the field. This reviewer agrees mostly with the interpretations and analysis provided in the manuscript for the first two. However, their interpretation or speculation on the third result is not convincing. The manuscript is quite hard to follow with numerous errors, inconsistencies, and inefficient presentation of information. Therefore, it is recommended that this manuscript should be resubmitted after a major revision.

Px Ly indicates page x and the y-th line from the top.

Suggestions/Corrections:

- Please provide clear (internal) structure of the cell. A diagram of the cut-out view of the cell, showing the channel and lead-in channel along with the position and orientation of the QTF's, would be helpful. The five images used for the cell (in the main text and Supplemental Information) are not efficiently chosen or presented.
- The figure numbers referred in the text are not correct on P11 and beyond.
- Please label the panels of Figure 5. In the caption, the 2nd sentence should be rewritten. In the 3rd sentence, "blue" is missing before (0 bar).
- What is the volume of HEC? Only the volume of IC is provided.
- It is quite difficult to follow the argument because there is lack of flow of discussion.

Questions:

On the experimental technique

It is fully understandable why the current cell geometry and technique are employed. One inherent drawback is that the measurement includes the interface thermal resistance between the channel and the bulk as in a two-probe technique. For example if the interface is formed between the A and B phase, its interface resistance might be substantial. Please comment on this issue.

On the results in normal fluid

The effective mean free path in a narrow channel follows $1/T$ below about 100 mK as a result of interference between the inelastic and elastic surface scattering. This projects an almost temperature independent thermal conductivity in the temperature range. However, the data shows a crossover to a different temperature dependence ($T^{-2.5}$) above around 20 mK. It is argued that the parallel conduction through the Kapitza resistance the IC body becomes dominant in this temperature range.

- The reported value of the Kapitza resistance at 20 mK is $AR_{\{K\}} \approx 10^{\{4\}} \text{ m}^{\{2\}} \text{ KW}^{\{-1\}}$ where A is the area of the boundary [Anders and Sprenger, in Proc. 14th Int. Conf. Low Temp. Phys. vol.1]. For this to be comparable to the channel thermal resistance, $R_{\{th\}} \approx 10^{\{6\}} \text{ KW}^{\{-1\}}$, the area should be of the order of $100 \text{ cm}^{\{2\}}$ in IC of $0.14 \text{ cm}^{\{3\}}$ volume. This seems to be quite high.
- It is known that the Kapitza resistance follows the acoustic mismatch model $R_{\{K\}} \sim T^{\{-3\}}$ for $T > 10 \text{ mK}$ and the temperature dependence weakens substantially below. Is this effect included in the theoretical curves?
- Figure 5: Are the theoretical curves for 0 bar or 22 bar? In the text (P14), it says the curve is for 22 bar.

On the anomalous response in superfluid

P6 L3: The length of the channel is 100 μm which is not much longer than the inelastic mean free path in superfluid. As shown in Figure 8, the mean free path in superfluid, in particular at low pressures, is comparable to the channel length.

- It might be possible that a portion of thermal quasiparticles in IC can ballistically travel to HEC and get registered by the QTF. This process does not require fountain pressure nor the aid of surface excitations.

- Reference 23 reports the formation of 2D domain walls in the B-phase, which probably disrupts the continuous formation of surface bound states on the bounding walls. If the surface excitations play an important role, what do you expect in the presence of the domain walls?
- Figure S2 shows the responses of the QTF for the reverse case of heat injection to HEC. Unlike the other case, the response between the two QTF seems to be reversed in the sense that peaks in IC (response) is larger than those in HEC (injection). Is there any explanation for this? For this it is important to know the exact placement of the QTF in each cell such as the distance from the channel opening and the orientation.

Reviewer #3 (Remarks to the Author):

In this paper, the authors report on the thermal transport of Helium-3 in a strongly confining channel. They study thermal transport through a microfabricated cavity in both the normal and superfluid ^3He , where the height of the cavity is comparable to the superfluid coherence length. In such a strongly confined geometry, there are several theoretical predictions on exotic quantum states associated with nontrivial topology and broken symmetries. This includes the thermodynamic stability of the stripe phase that spontaneously breaks translational symmetry along the confinement, and the presence of helical Majorana fermions that are topologically protected gapless surface Andreev bound states. Both are reflection of unconventional and topological superfluidity of liquid ^3He .

I think the following new findings reported in the manuscript are enough to warrant publication in Nature Communications: (i) The thermal conductivity of the normal ^3He measured is independent of temperatures below 10mK at both 0 bar and 22 bar. (ii) Measurements of thermal conductivity in the superfluid phase show weak T-dependence, which is compared with the theory for bulk diffusive thermal conductivity. As the strong confinement in a microfabricated channel is enough to clamp the normal component, the channel is well in the Knudsen regime and the contribution from hydrodynamic flow is negligible. (iii) The anomalous thermal response in the HEC fork is reported, which cannot be accounted for by the hydrodynamic heat flux and indicates a ballistic flow of quasiparticles generated in the IC. All the results in the superfluid state well motivates the development of theory to incorporate surface bound states under nonequilibrium conditions.

Recently, the general concept of topology and symmetry is expanding and penetrating to the fields of condensed matter physics. Studies of thermal transport in a confined geometry may provide a powerful way for capturing the direct signatures of topological quasiparticles. I think, therefore, main observations reported in the manuscript impact audiences in the diverse fields of condensed matter physics. In summary, there are new and interesting results presented, the manuscript should be published in Nature Communications.

My minor comments are listed below:

-- In page 17, the authors speculate that surface excitations respond to the fountain pressure between two channels and may flow with little dissipation. This might be one possible scenario. However, the diffusive scattering of quasiparticles on rough surface may give rise to dissipative transport through the channel unless the wall is coated by ^4He layers. Can you make a brief comment about the effect of surface roughness on ballistic thermal transport in the channel.

-- There are typos in Figure number referred in the main text, where "Figs.6-8" are referred to as "Fig.5" in pages 11-16. Please check consistency and fix properly.

-- I recommend to add the following paper in the reference: Y. Nagato et al., JLTP 110, 1135 (1998). This should be cited together with Refs.45-47.

The authors would like to thank all three reviewers for their numerous detailed comments and suggestions. We have responded to the comments as noted below. We apologize for the several typographical errors in the previous version and have worked diligently to address these. Below, we address the reviewers' comments in black and our responses in blue:

Reviewer 1

1. All figures, both schematics and graphs, could be greatly improved with consistent style, labelling and keys.

This has been done. We split the two pressures apart in Figs 3, 4 aiding the clarity. Fig 5 was also redrawn with altered colors. We added an inset to figure 6 and changed the colors to be consistent with the colors in Figs. 7 and 8. The same scheme is carried forward in supplemental figures S 2, S 3, and S 4.

2. I do not agree with the statement on Page 13 that the data in Figs 3(b) and 4 clearly lie between the extremes.

The 22 bar data appear to follow the point scatterer theory lines very well below 10mK. This should be noted, commented and compared.

We agree with the referee's comment and have revised the text. The discussion relating to the results in the normal state has been extensively rewritten. On Page 12 1st complete para we have restated the description of the figure, and in the 1st para on page 12 we state that the crossover at 22 bar should occur below ~ 2.6 mK (near T_c).

3. If possible I would like to see more argument made for the existence of transport due to surface excitations. In my view this is the most important and provocative result of this work. Augmenting this aspect may require a re-structure to reduce the focus on the normal state measurements.

We have rewritten the introductory section, the last paragraph now explicitly states that the excitations incident on the HEC fork must be flowing ballistically and make reference to the recent results of Autti *et al.* (added Ref 60). We have also restructured the discussion section extensively. We make the argument explicitly that the anomalous excitations originate in the channel or its ends. We also argue that these excitations are unlikely to be bulk quasiparticles as the distance between the channel and the HEC fork is much greater than the quasiparticle mean free path (page 18 1st and 2nd paragraph).

4. Attempt to make clearer, more quantitative statements of how helium-3 in this slab geometry can be a model for other quantum matter systems, particularly superconducting ones.

We have restructured the first 3 paragraphs in the introduction to address this point.

Reviewer 2 provided a strong critique which we have addressed by restructuring the paper and have addressed also the specific comment in the revised text.

“However, their interpretation or speculation on the third result (anomalous conduction in the superfluid phase) is not convincing. The manuscript is quite hard to follow with numerous errors, inconsistencies, and inefficient presentation of information.”

We have rewritten the discussion section (pages 18-19-20), with an increased focus on the potential mechanism for anomalous conduction. We have expanded and clarified our proposal that surface excitations are responsible. We believe the recent independent observation of long lived transport traceable to surface excitations by Autti et al (newly added Ref 60), tends to add support to this model. We rule out an alternative possibility of fountain flow induced superfluid deficit, adding a discussion in Supplementary Note S-2.

Suggestions/Corrections

1. Please provide clear (internal) structure of the cell. A diagram of the cut-out view of the cell, showing the channel and lead-in channel along with the position and orientation of the QTF's, would be helpful. The five images used for the cell (in the main text and Supplemental Information) are not efficiently chosen or presented.

This is now done, please see Figure 1. The clarification of this geometry is important to rule out ballistic propagation of bulk thermal quasiparticles, see below.

2. The figure numbers referred in the text are not correct on P11 and beyond.

We apologize. This is now fixed

3. Please label the panels of Figure 5. In the caption, the 2nd sentence should be rewritten. In the 3rd sentence, “blue” is missing before (0 bar).

We apologize. This is now fixed

4. What is the volume of HEC? Only the volume of IC is provided.

We have added this useful information, please see paragraph 2 on Pages 8-9

It is quite difficult to follow the argument because there is lack of flow of discussion.

We agree with the referee's comment. We have redrafted and added material to the discussion section, as discussed elsewhere.

Questions:

On the experimental technique

It is fully understandable why the current cell geometry and technique are employed. One inherent drawback is that the measurement includes the interface thermal resistance between the channel and the bulk as in a two probe technique. For example if the interface is formed between the A and B phase, its interface resistance might be substantial. Please comment on this issue.

We agree with the referee that in principle there should be an effect due to Andreev scattering at the A-B interfaces. However, the A phase in the channel would be oriented so that the gap would be largest in the direction perpendicular to heat flow, making the differences in the gap small. In any event, we cannot resolve any additional boundary resistance when there is A in the channel and B in the bulk regions. A statement to this effect is included in the last complete paragraph on page 13.

On the results in normal fluid

The effective mean free path in a narrow channel follows $1/T$ below about 100 mK as a result of interference between the inelastic and elastic surface scattering. This projects an almost temperature independent thermal conductivity in the temperature range. However, the data shows a crossover to a different temperature dependence ($T^{-2.5}$) above around 20 mK. It is argued that the parallel conduction through the Kapitza resistance the IC body becomes dominant in this temperature range.

1. The reported value of the Kapitza resistance at 20 mK is $AR_{\{K\}} \approx 10^{\{4\}} \text{ m}^{\{2\}} \text{KW}^{\{-1\}}$ where A is the area of the boundary [Anders and Sprenger, in Proc. 14th Int. Conf. Low Temp. Phys. vol.1]. For this to be comparable to the channel thermal resistance, $R_{\{th\}} \approx 10^{\{6\}} \text{KW}^{\{-1\}}$, the area should be of the order of $100 \text{ cm}^{\{2\}}$ in IC of $0.14 \text{ cm}^{\{3\}}$ volume. This seems to be quite high.

We have now addressed this issue and the issue raised in the next point under Methods, section titled “Calculation of thermal resistance” on p25-26. In following up the referee’s comments, we realized that the scaling of the thermal resistance accounting for the difference in sound velocity and density between mixtures and pure ^3He had not been done (though we did scale with sound velocity and density with pressure). We also outline why we prefer to use the Lancaster results over the Andres/Sprenger result for sinter. See also new supplemental figures S 6, S 7. In either case, the point is that below 10 mK where the conduction through the channel dominates there is no significant difference. We are happy to use the A/S result if the referee believes it to be more pertinent.

2. It is known that the Kapitza resistance follows the acoustic mismatch model $R_{\{K\}} \sim T^{\{-3\}}$ for $T > 10$ mK and the temperature dependence weakens substantially below. Is this effect included in the theoretical curves?

Yes please see discussion on page 25-26 see also Supplemental Figures S 6, S 7.

3. Figure 5: Are the theoretical curves for 0 bar or 22 bar? In the text (P14), it says the curve is for 22 bar.

We have now explicitly stated that the curve is for 21 bar (page 13 paragraph starting “the measured relaxation times..”, and thank the referee for pointing this out. Since the thermal conductivity is not scaled to that at T_c , it would be natural to associate the curves with 0 bar data. To distinguish this, we show the theory curves in red in Fig 5 to emphasize the association with 22 bar data.

On the anomalous response in superfluid

P6 L3: The length of the channel is 100 μm which is not much longer than the inelastic mean free path in superfluid. As shown in Figure 8, the mean free path in superfluid, in particular at low pressures, is comparable to the channel length.

1. It might be possible that a portion of thermal quasiparticles in IC can ballistically travel to HEC and get registered by the QTF. This process does not require fountain pressure nor the aid of surface excitations.

We have clarified the internal structure of the cell. This shows that the relevant lengths are much longer than the channel length, and very long compared to the mean free path. Also now noted in the discussion section (see the first and second paragraph on page 18). Transmission via bulk quasiparticles would require a very long tail in the distribution. Also, it would still be observable in the normal state at zero bar and at 0.62 bar – it is not.

Please see the following discussion on page 18. We also note the result by Autti et al [Ref 60] is of relevance here, where the transmission of surface bound excitations through the bulk is noted.

2. Reference 23 reports the formation of 2D domain walls in the B-phase, which probably disrupts the continuous formation of surface bound states on the bounding walls. If the surface excitations play an important role, what do you expect in the presence of the domain walls?

Hard to say – but the point is well taken. The B phase in the channel would be present at low temperatures, where the effect is strongest. The presence of a modulated phase in principle should influence the nature of the surface bound excitations (now noted at the bottom of page 19 and top of page 20). In contrast to the results in ref 8, which were done with $64 \mu\text{mol}/\text{m}^2$ ^4He pre-plating (5 layers of ^4He equivalent and thus specular scattering), our experiments reported here were done with essentially pure ^3He . We do not identify any specific feature with this region of temperature and pressure.

3. Figure S2 shows the responses of the QTF for the reverse case of heat injection to HEC. Unlike the other case, the response between the two QTF seems to be reversed in the sense that peaks in IC (response) is larger than those in HEC (injection). Is there any explanation for this? For this it is important to know the exact placement of the QTF in each cell such as the distance from the channel opening and the orientation.

We note that in the scenario that we think is most likely, the surface bound excitations would be injected into the bulk from the region at the cold end of the channel where the inflow of superfluid is highest. Thus the fact that the IC fork is located much closer to the outlet and in line with the outlet enhances the observation of the reverse anomalous effect. We have added a statement at the top of Page 17. The geometry of the cell and placement of the QTFs is clarified in Fig. 1.

Reviewer #3

1. In page 17, the authors speculate that surface excitations respond to the fountain pressure between two channels and may flow with little dissipation. This might be one possible scenario. However, the diffusive scattering of quasiparticles on rough surface may give rise to dissipative transport through the channel unless the wall is coated by ^4He layers. Can you make a brief comment about the effect of surface roughness on ballistic thermal transport in the channel.

It is difficult to do this since we did not perform experiments with various surface coatings of ^4He . The fact that the surface is covered by localised ^3He implies that the scattering is stronger than a purely diffuse scattering as we now note on page 19-20. However, we note that the effect must be associated with the superfluid state in the channel and long mean free paths. Thus, purely diffusive transport can be ruled out. We have included a discussion on page 15 and also Supplemental Note S-2.

2. There are typos in Figure number referred in the main text, where "Figs.6-8" are referred to as "Fig.5" in pages 11-16. Please check consistency and fix properly.

We apologize for this oversight. It has now been addressed.

3. I recommend to add the following paper in the reference: Y. Nagato et al., JLTP 110, 1135 (1998). This should be cited together with Refs.45-47.

We have added the reference.

We agree with the referee that “all the results in the superfluid state well motivates the development of theory to incorporate surface bound states under nonequilibrium conditions”.

We hope that our substantially revised manuscript better demonstrates the wealth of new phenomena, and expect it to stimulate further experimental and theoretical work, as well as better highlighting their broader relevance.

REVIEWERS' COMMENTS:

Reviewer #1 (Remarks to the Author):

The authors are to be commended on the significant improvements that have been made to this manuscript, better reflecting their scientific work of high quality. All my reservations and comments have been addressed satisfactorily.

Reviewer #2 (Remarks to the Author):

The authors has made significant revisions and answered all three reviewers' questions and comments in detail. The manuscript is in much better shape. Allowing some differences in specific interpretation of the results, it reports a high quality results from quite challenging experiments. The main result of this work is a new mechanism of thermal conduction through surface excitations in a confined superfluid, which is very intriguing not only in this superfluid ^3He system but also in other topological materials where the edge states play a crucial role in various transport phenomena. The data analysis is thorough, and the authors laid out their interpretation with strong supporting analysis and argument. I am satisfied with the revised manuscript and recommend publication in Nature Communications

Reviewer #3 (Remarks to the Author):

The authors have convincingly answered to all of my previous comments, and the manuscript has been well improved by the authors. The improved manuscript would remove potential confusion of the readers. I believe that the manuscript triggers an important step for understanding the roles of low-lying quasiparticles on anomalous transport phenomena in topological superconductors. Hence, I feel that the manuscript is now suitable for publication in Nat. Commun.

The Reviewers comments (below) communicated to us on 8/22/2020 indicate that no further revisions are needed. This page is included as a formal part of the resubmission process. The authors would like to express their thanks to the reviewers for their constructive comments throughout this process.

Jeevak Parpia

For authors.

REVIEWERS' COMMENTS:

Reviewer #1 (Remarks to the Author):

The authors are to be commended on the significant improvements that have been made to this manuscript, better reflecting their scientific work of high quality. All my reservations and comments have been addressed satisfactorily.

Reviewer #2 (Remarks to the Author):

The authors has made significant revisions and answered all three reviewers' questions and comments in detail. The manuscript is in much better shape. Allowing some differences in specific interpretation of the results, it reports a high quality results from quite challenging experiments. The main result of this work is a new mechanism of thermal conduction through surface excitations in a confined superfluid, which is very intriguing not only in this superfluid ^3He system but also in other topological materials where the edge states play a crucial role in various transport phenomena. The data analysis is thorough, and the authors laid out their interpretation with strong supporting analysis and argument. I am satisfied with the revised manuscript and recommend publication in Nature Communications

Reviewer #3 (Remarks to the Author):

The authors have convincingly answered to all of my previous comments, and the manuscript has been well improved by the authors. The improved manuscript would remove potential confusion of the readers. I believe that the manuscript triggers an important step for understanding the roles of low-lying quasiparticles on anomalous transport phenomena in topological superconductors. Hence, I feel that the manuscript is now suitable for publication in Nat. Commun.